# A Single Session of Beach Tennis with Recreational Athletes Improves Anxiety Symptoms in Women but Not in Men: A Randomized Trial

**DOI:** 10.3390/ijerph22010038

**Published:** 2024-12-30

**Authors:** João Victor Rosa de Freitas, Bráulio Evangelista de Lima, Rizia Rocha-Silva, Vinnycius Nunes de Oliveira, Thalles Guillarducci Costa, Mila Alves Matos Rodrigues, Rodrigo Luiz Vancini, Marília Santos Andrade, Gustavo de Conti Teixeira Costa, Lorenzo Laporta, Ricardo Borges Viana, Claudio Andre Barbosa de Lira

**Affiliations:** 1Faculty of Physical Education and Dance, Federal University of Goiás, Goiânia CEP 74690-900, Brazil; joao.rosa@discente.ufg.br (J.V.R.d.F.); brauliofisio@discente.ufg.br (B.E.d.L.); vinnyciusnunes@discente.ufg.br (V.N.d.O.); milaalves@discente.ufg.br (M.A.M.R.); conti02@ufg.br (G.d.C.T.C.); 2Center for Teaching and Research Applied to Education, Federal University of Goiás, Goiânia CEP 74690-900, Brazil; rizia.rocha@ufg.br; 3University Unit of Itumbiara, State University of Goiás, Itumbiara CEP 75536-100, Brazil; thalles.costa@ueg.br; 4Department of Sports, Center for Physical Education and Sports, Federal University of Espírito Santo, Vitória CEP 29075-010, Brazil; rodrigoluizvancini@gmail.com; 5Department of Physiology, Federal University of São Paulo, São Paulo CEP 04023-062, Brazil; marilia1707@gmail.com; 6Sports Performance Analysis Study Center, Federal University of Santa Maria, Santa Maria CEP 97105-900, Brazil; laportalorenzo@gmail.com; 7Institute of Physical Education and Sports, Federal University of Ceará, Fortaleza CEP 60455-760, Brazil; vianaricardoborges@ufc.br

**Keywords:** beach tennis, mental health, physical exercise, racket sports

## Abstract

Introduction: Beach tennis has become a popular sport, but research on its mental health benefits is scarce. To the best of our knowledge, no studies have examined the effects of beach tennis on anxiety symptoms. Objectives: To assess the effect of a single session of beach tennis, in both singles and doubles modes, on anxiety symptoms. Methods: Twenty-two recreational players (11 women, age: 35.00 [13.50] years) were evaluated. Each participant underwent three intervention sessions in random order: one-on-one match (singles), two-on-two match (doubles), and a control session (non-exercise). State anxiety and affective responses were measured before and after each intervention. Results: For men, no significant interaction between time and session was found (*p* = 0.646). In women, there was a significant interaction between time and session (*p* = 0.002). Anxiety symptoms significantly decreased from pre- to post-singles condition (*p* = 0.007) and from pre- to post-doubles condition (*p* = 0.010). A significant difference was observed between the post-singles and post-control conditions (*p* = 0.002). Conclusion: Beach tennis demonstrated an anxiolytic effect in women, with significant reductions in anxiety symptoms following both singles and doubles sessions. However, no such effects were observed in men. These findings suggest that beach tennis could be a viable intervention for managing anxiety symptoms in women. The study was registered in the Brazilian Clinical Trials Registry (registration code: U1111-1293-7949).

## 1. Introduction

Physical exercise is widely recognized for its positive impact on physical fitness and its role as a non-pharmacological tool for preventing and managing non-communicable diseases, including cardiovascular, endocrine, musculoskeletal, respiratory, neurological, and psychiatric diseases [1]. Anxiety disorders, in particular, represent a significant public health concern [2,3]. According to the World Health Organization (WHO), 264 million people were affected by anxiety disorders in 2015, reflecting a global prevalence of 3.6% [4]. This number rose to 374 million in 2020 due to the COVID-19 pandemic [5].

Young adults, in particular, experience higher rates of anxiety [6], with women disproportionately affected compared to men due to a combination of biological, social, and cultural factors [4,7]. Indeed, women tend to report higher levels of both state and trait anxiety compared to men due to social, cultural, and biological factors, as well as differences in physiological stress responses [8,9]. For instance, women report higher levels of both state and trait anxiety, potentially influenced by physiological differences, such as hormonal fluctuations associated with the menstrual cycle, which may exacerbate anxiety symptoms [10]. These findings underscore the importance of exploring and developing tailored treatments for anxiety, with a particular focus on addressing the unique needs of women.

A substantial body of research demonstrates that both acute and chronic physical exercise can be protective against psychiatric symptoms and can effectively reduce these symptoms in individuals with and without anxiety disorders [11]. For instance, Verhoeven et al. [12]. compared the effects of antidepressants and running therapy on mental and physical health outcomes. In their study, 45 participants received antidepressant medication, while 96 underwent a running therapy program consisting of 45 min outdoor sessions performed two to three times per week at an intensity corresponding to 70–85% of their heart rate reserve. While no significant differences in mental health outcomes were observed between the two groups, running therapy resulted in significantly better physical health outcomes compared to antidepressants. Henriksson et al. [13] investigated the effects of an exercise intervention on anxiety symptoms and evaluated the efficacy of moderate/high-intensity exercise versus low-intensity exercise in primary care patients diagnosed with anxiety disorders. The study included 286 patients randomly assigned to one of three groups: a low-intensity exercise program, a moderate/high-intensity exercise program, or a control group receiving standard non-exercise treatment. Both exercise interventions lasted 12 weeks, with three sessions per week. The results showed that both exercise programs significantly reduced anxiety symptoms, with moderate/high-intensity exercise yielding the most pronounced benefits.

Despite the well-documented benefits of physical exercise for managing anxiety, 31% of the world’s adult population—approximately 1.8 billion individuals—remains physically inactive [14]. Common barriers to physical activity include lack of time, energy, social support, motivation, and perceptions of exercise as boring [15]. Therefore, developing more attractive, innovative, and enjoyable types of physical exercise may be crucial for increasing population engagement and reaping the mental health benefits associated with physical activity.

Studies on innovative forms of exercise have yielded promising outcomes. For instance, exergames have been reported to be more enjoyable than traditional exercises such as walking or running and are also more effective in reducing anxiety symptoms [9]. Similarly, Santos et al. [16] found that high-intensity functional training was perceived as more enjoyable than running. These findings are significant, as a common reason for avoiding exercise is that people find it boring and unattractive [15]. Consequently, new exercise modalities may reduce physical inactivity by offering more enjoyable alternatives.

Beach tennis is an emerging sport involving two teams—comprising one or two players—competing on a sand court. Despite its growing popularity, only nine studies related to beach tennis have been published in journals indexed in PubMed (as of 25 November 2024). This number is relatively small compared to other sports, such as “tennis” (10,644 articles), “football” (19,285 articles), and “volleyball” (2686 articles), based on searches using MeSH terms. Thus, additional research is needed to further characterize the sport.

To the best of our knowledge, only Carpes et al. [17] investigated enjoyment in beach tennis, reporting high levels of enjoyment among practitioners. However, that study focused on individuals with arterial hypertension. Furthermore, no study investigated the effect of beach tennis on anxiety symptoms. Therefore, research evaluating enjoyment in a non-clinical population and changes in anxiety symptoms due to beach tennis practice is necessary.

Regarding racket sports and anxiety symptoms, Conde-Ripoll et al. [18] analyzed pre- and post-competitive anxiety and self-confidence in relation to match outcomes among 11 high-level male padel players from Finland. The study found that losing players experienced significant increases in cognitive, somatic, and state anxiety, along with a notable decline in self-confidence, while winning players showed only a significant increase in state anxiety. The authors concluded that players should develop mental skills to manage errors and defeats, and coaches are encouraged to include pressure training and foster a winning mentality during simulated matches. However, this study did not include female athletes.

Considering the rising number of beach tennis practitioners, the limited research on this sport, the high prevalence of anxiety disorders (particularly among women) [7], and the role of motivation in physical activity adherence, a comprehensive investigation is warranted. Additionally, there is a lack of studies characterizing the intensity of beach tennis sessions. This study aimed to evaluate and compare the acute effects of a beach tennis session on anxiety symptoms in women and men. Secondary objectives included characterizing enjoyment, affective responses, and exercise intensity (measured by heart rate) during the session. We hypothesized that beach tennis would have an anxiolytic effect, driven by high levels of enjoyment and positive affective responses. We also expected the intensity of the exercise to be classified as moderate.

## 2. Materials and Methods

### 2.1. Participants

Thirty beach tennis players (16 women and 14 men) were invited to participate in the study. Participants were recruited through direct invitations at beach tennis academies in Goiânia, Brazil, as well as through posts on the researchers’ social media accounts (Instagram, Facebook, and X [formerly Twitter]).

The inclusion criteria required participants to (a) have no contraindications for physical activity, as assessed by the Physical Activity Readiness Questionnaire (PAR-Q) [13]; (b) be aged 18 to 40 years, a range selected due to the high prevalence of anxiety symptoms in this age group, particularly among women; (c) be recreational beach tennis players with at least 2 months of experience in beach tennis, ensuring a basic familiarity with the sport; and (d) be literate, to fully understand the informed consent and study protocols. Participants were excluded if they (a) had osteomuscular injuries, that could impair their performance or compromise safety. All participants provided written informed consent after receiving detailed information about the study procedures. The study was approved by the Research Ethics Committee of the Federal University of Goiás (CAAE 67075723.8.0000.5083) and registered in the Brazilian Clinical Trials Registry (registration code: U1111-1293-7949).

### 2.2. Experimental Design

This study used a randomized crossover clinical trial conducted across four sessions, with intervals of 24–72 h between sessions (Figure 1).

In the first session, participants completed an anamnesis to screen for contraindications to physical activity, followed by assessments of anthropometric data, anxiety traits, and cardiorespiratory fitness. At the end of the session, participants were randomized using web-based software (www.randomizer.org) to determine the intervention sequence (control, single session, and double session) for the subsequent sessions.

The interventions in sessions 2, 3, and 4 consisted of sessions conducted under three conditions: (a) single session: one-on-one matches, (b) double session: two-on-two matches, and (c) control session: non-exercise. State anxiety symptoms and affective responses were measured immediately before and after each intervention. Enjoyment was assessed only immediately after the beach tennis sessions, and heart rate was recorded throughout all interventions. The outcome measures employed in this study were selected due to their robust validation, cost-effectiveness, and prior cultural adaptation to Brazilian Portuguese, ensuring reliable and accurate assessment within the studied population.

The decision to evaluate both single and double matches was based on the fact that these modalities are commonly used in training and competitions. Therefore, it is essential to conduct scientific studies that investigate potential differences in physiological and psychological responses between these two forms of play.

Participants were requested to wear appropriate gear for physical activity. Stimulants, such as caffeine or tea, were avoided for at least 8 h before each visit. Furthermore, participants were instructed to avoid eating within 2 h before each visit and to follow their regular hydration routine. No training sessions were allowed on the day of laboratory visits, and participants were instructed to refrain from intense physical exercise for at least 24 h prior to each visit.

No medical complications occurred during the experimental procedures.

### 2.3. Experimental Procedures

#### 2.3.1. Anamnesis and Anthropometric Evaluation Assessment

The PAR-Q [19] was administered to assess participants’ general health. Anthropometric measurements included body mass (measured using an Omron HN-289 digital scale, Hoffman Estates, IL, USA) and height (measured using a stadiometer, Caumaq, Cachoeira do Sul, Brazil). Body mass index (BMI) was calculated using the following formula: BMI = body mass (kg)/height^2^ (m) [20].

#### 2.3.2. Cardiorespiratory Fitness Assessment

Participants performed a cardiopulmonary exercise test to determine maximal oxygen uptake ($\dot{V}O_{2}$max) and maximum heart rate (HRmax). Testing was conducted using the VO2000 metabolic system (Medical Graphics Corporation, Saint Paul, MN, USA) and followed the manufacturer’s calibration protocols. Participants wore a mouthpiece connected to the metabolic system with a nasal clip to prevent nasal airflow [21]. Additionally, a heart rate monitor (model H7, Polar, Kempele, Finland) was positioned at the xiphoid process.

The test was performed on a treadmill (model ATL, Inbramed, Porto Alegre, Brazil) and included a 5 min warm-up at 6 km/h, followed by an initial speed of 7 km/h, which increased by 1 km/h per minute until exhaustion [22]. To determine $\dot{V}O_{2}$max, data were filtered by taking a 10-s average of the measurements. $\dot{V}O_{2}$max was identified at the point where a further increase in workload no longer resulted in $\dot{V}O_{2}$elevation, indicating a plateau [23]. HRmax was recorded as the highest heart rate achieved during the test.

#### 2.3.3. State-Trait Anxiety Assessment

The State-Trait Anxiety Inventory (STAI) was used to assess state anxiety symptoms. The validated Brazilian Portuguese version of the inventory was utilized [24]. The STAI is considered the gold standard for assessing anxiety and is widely used. It consists of two parts: state and trait anxiety.

In this study, the state scale was used to evaluate participants’ state anxiety symptoms before and after the interventions, while the trait scale was used to characterize the participants and assessed participants’ general emotional tendencies. Both parts contain 20 statements, each reported on a 4-point Likert scale. For trait anxiety, participants responded with 1 = almost never, 2 = sometimes, 3 = often, and 4 = almost always. For state anxiety, responses were 1 = not at all, 2 = somewhat, 3 = moderately, and 4 = very much.

STAI scores range from 20 to 80 points, with higher scores indicating higher anxiety levels [24]. The following cutoff points were used: ≤50 for low anxiety and >50 for high anxiety [25].

The internal consistency of the STAI is strong, with Cronbach’s alpha values ranging from 0.86 to 0.95 [25]. The trait anxiety inventory was administered by a trained researcher during the participants’ first visit and was used to characterize them. The state anxiety inventory was also administered by a trained researcher during subsequent visits at two time points: immediately before and after conditions (single session, double session, and control session).

#### 2.3.4. Enjoyment Assessment

Enjoyment was assessed using the Physical Activity Enjoyment Scale (PACES) [26]. For this purpose, the validated Brazilian Portuguese version of the scale was utilized [27]. PACES consists of 18 items, each containing two opposing statements separated by a 7-point Likert scale (1 = I enjoy it, 4 = neutral, and 7 = I hate it), with a total score ranging from 18 to 126 points. Higher scores indicate greater enjoyment of the physical activity [27]. Participants responded to the scale after hearing the following instruction: “Please evaluate how you felt during the exercise you just performed.”

The intraclass correlation coefficient (ICC) was classified as almost perfect (0.910) [27]. The scale was administered immediately after the beach tennis sessions. It was not applied during the control session, as this was not a physical activity scenario, and the instrument is intended to assess enjoyment specific to exercise or sport [26].

#### 2.3.5. Affective Response Assessment

Affective responses were measured using the affective valence scale developed by Hardy and Rejeski [28], which was translated and adapted into Brazilian Portuguese by Alves et al. [27]. This scale ranges from +5 to −5, representing varying degrees of feelings during physical activity: +5 = very good, +3 = good, +1 = fairly good, 0 = neutral, −1 = fairly bad, −3 = bad, and −5 = very bad. The ICC was classified as large (0.644) [27]. The scale was presented to participants immediately before and after each condition (single session, double session, and control session).

#### 2.3.6. Heart Rate Assessment

Heart rate was monitored using a heart rate monitor (model H7, Polar, Kempele, Finland) placed at the level of the xiphoid process. Heart rate measurements were recorded every 5 s throughout the beach tennis sessions. After each session, the data were transferred to the Polar Flow software (version 6.23.0, Polar, Kempele, Finland) and extracted to a Microsoft Excel file (Microsoft Excel^®^ for Microsoft 365, Washington, DC, USA).

Heart rate values were classified according to the criteria established by the American College of Sports Medicine [29]. Specifically, light exercise intensity corresponds to 57–63% of maximal heart rate, moderate intensity to 64–76% of maximal heart rate, vigorous intensity to 77–95% of maximal heart rate, and near-maximal to maximal intensity to ≥96% of maximal heart rate.

#### 2.3.7. Temperature and Relative Humidity Assessment

Temperature and relative humidity were recorded using a digital thermo-hygrometer (Simpla–TH01, São Leopoldo, Brazil) prior to the beach tennis sessions. During the single session, the median temperature was 24.9 °C (IQR, 3.4 °C; min–max, 21.0 °C–31.8 °C), with a median relative humidity of 51.0% (IQR, 17.7%; min–max, 35.1–72.0%). For the double session, the median temperature was 23.2 °C (IQR, 2.8 °C; min–max, 21.0 °C–32.0 °C), and the median relative humidity was 58.0% (IQR, 27.7%; min–max, 35.0–85.0%). There were no statistically significant differences in temperature (*p* = 0.251) or humidity (*p* = 0.058) between the sessions.

#### 2.3.8. Beach Tennis Sessions

The beach tennis protocol was adapted from Carpes et al. [17], Domingues et al. [30], and Ferrari et al. [31]. Each session began with a 5 min warm-up comprising technical drills, including volleys, short balls, smashes, hooks, lobs, and serves. Participants then played three 15 min matches, with 2 min passive recovery intervals between each match, resulting in a total session time of 56 min. Singles and doubles sessions were conducted on different days.

The court dimensions were standardized: 16 × 8 m for doubles matches and 16 × 4.5 m for singles matches. The net height was set at 1.70 m for both men and women. The net height was measured using an aluminum net height stick before each session. A uniform net height was maintained for men and women, as beach tennis classes in academies are traditionally conducted with mixed-gender participants. Furthermore, the studies referenced in this research applied the same net height [17,30]. The court surface consisted of medium-grain white sand, carefully leveled before each session to ensure a flat, uniform surface free of stones and other irregular objects.

A familiarization session was deemed unnecessary, as all participants had prior experience with the sport. Two certified members of the research team, holding credentials from the Brazilian Tennis Confederation, evaluated each participant and classified them into one of two categories: intermediate (demonstrating agility and coordinated movements) or advanced (demonstrating mastery of all game aspects). Classification criteria included experience, competition participation, competition category, and the coach’s evaluation. This categorization ensured players were matched with others of similar skill levels, enhancing the quality of the matches. Additionally, sessions were organized by gender to foster appropriate motivation and intensity. Athletes were verbally encouraged throughout the matches to perform optimally.

#### 2.3.9. Control Session

The control session consisted of 56 min of seated rest without any physical activity. Participants were prohibited from using smartphones, communicating, or sleeping during this session. The control session was conducted in a temperature-controlled room (23 °C) within the beach tennis club. Anxiety symptoms and affective responses were measured as described for the beach tennis sessions.

### 2.4. Statistical Analysis

Data analysis was performed using the Jeffreys’s Amazing Statistics Program (JASP, version 0.16.3, University of Amsterdam, The Netherlands). The Shapiro–Wilk test was used to assess data normality, and Mauchly’s test examined the sphericity of residuals. Both tests were supplemented with visual inspections (histograms and Q–Q plots). Levene’s test evaluated the homogeneity of variances.

Normally distributed data are reported as mean ± standard deviation (SD), while non-normally distributed data are presented as the median and interquartile range (IQR). Categorical data are expressed as absolute and relative frequencies. For comparisons, mean differences with 95% confidence intervals (95% CIs) were calculated for parametric tests, while Hodges–Lehmann estimates were used for non-parametric tests.

Due to differences in anxiety prevalence between genders, separate analyses were conducted for men and women. Research suggests that social and biological factors contribute to a higher likelihood of anxiety symptoms among women [32,33], highlighting the importance of gender-specific analyses to prevent biases stemming from uneven representation in sports and exercise studies.

Consequently, we conducted separate analyses by gender, as applying the same comparison parameters would be inappropriate given the higher prevalence of anxiety among women. This approach ensures a more precise and representative evaluation of the data.

To assess the acute effect of beach tennis on state anxiety levels, we performed a repeated-measures analysis of variance (ANOVA) to analyze the interaction between conditions (single session × double session × control session) and time (pre-game × post-game). Post hoc Conover analysis with Bonferroni correction was conducted to identify significant differences between the intervention pairs. To meet the assumptions for repeated-measures ANOVA, we transformed the state anxiety variable using its inverse.

A paired Student’s *t*-test was used to compare the mean heart rate of women across sessions (single and double) due to the parametric nature of the data. In contrast, the comparison of men’s heart rate across sessions was performed using the Wilcoxon test due to the non-parametric distribution of the data.

To evaluate the acute effect of beach tennis on enjoyment, paired Student’s *t*-tests were conducted separately for women and men, focusing on post-session enjoyment (single session × double session). The Friedman test analyzed both genders, examining interactions between sessions (single, double, and control) and time (pre-session and post-session). Post hoc Conover analysis with Bonferroni correction identified differences between the interventions.

Effect sizes were calculated for each statistical test. Cohen’s *d* was used for the paired *t*-test and for the post hoc test for repeated-measures ANOVA, classified as “trivial” (*d* < 0.2), “small” (0.2 ≤ *d* < 0.5), “medium” (0.5 ≤ *d* < 0.8), and “large” (*d* ≥ 0.8) [34]. For the Wilcoxon test, we used the rank-biserial correlation coefficient (r_B_) [35], classified similarly to Pearson correlations: “trivial” (r_B_ < 0.10), “small” (0.10 ≤ r_B_ < 0.30), “medium” (0.30 ≤ r_B_ < 0.50), and “large” (r_B_ ≥ 0.50) [36]. Kendall’s *W* was used to measure effect size for the Friedman test, with values interpreted as “no agreement” (<0.10), “weak agreement” (0.10 ≥ *W* < 0.30), “moderate agreement” (0.30 ≥ *W* < 0.60), “strong agreement” (0.60 ≥ *W* < 1.0), and “perfect agreement” (*W* = 1) [37]. Omega squared (ω^2^) was used to measure effect size for repeated-measures ANOVA, classified as “trivial” (<0.01), “small” (0.01 ≤ ω^2^ < 0.06), “medium” (0.06 ≤ ω^2^ < 0.14), and “large” (ω^2^ ≥ 0.14) [32]. Statistical significance was set at *α* = 0.05.

## 3. Results

### 3.1. Participants’ Characteristics

Eight participants did not complete all experimental procedures, leaving a final sample of 22 participants (11 women and 11 men). Figure 2 illustrates the study flow, and Table 1 and Table 2 summarize the participants’ general characteristics and experience with beach tennis.

All participants underwent cardiopulmonary exercise testing under optimal technical conditions. The average $\dot{V}O_{2}$max for the sample was 44.95 [13.17] mL/kg/min, and the mean HRmax was 181 [7.00] bpm, corresponding to 99.1% of the predicted HRmax based on age. As expected, $\dot{V}O_{2}$max was higher in men than in women, while HRmax did not differ significantly between the sexes.

All participants met the physical activity guidelines, engaging in more than 150 min of physical activity per week. The median experience with beach tennis was 14.5 months. Additionally, 19 participants had competed in recreational tournaments, with an average of 3.2 competitions per participant. None of the participants reported injuries related to beach tennis practice (Table 2).

### 3.2. State Anxiety Symptoms

For women, repeated measures ANOVA did not show a significant session effect (*F*(2, 20) = 3.377, *p* = 0.054, *ω^2^* = 0.019). However, there was a significant time effect (*F*(1, 10) = 11.706, *p* = 0.007, *ω^2^* = 0.054, “small”) and a significant interaction between time and session (*F*(2, 20) = 9.059, *p* = 0.002, *ω^2^* = 0.034, “small”). Post hoc analysis with Bonferroni correction indicated a significant reduction in state anxiety symptoms after both the single session (*p* = 0.007, d = −0.744, “medium”) and the double session (*p* = 0.010, d = −0.720, “medium”); however, no change was observed after the control session (*p* = 1.000). State anxiety was also lower following the single session compared to the control session (*p* = 0.002, d = 0.777, “medium”, Figure 3 and Table 3).

For men, repeated-measures ANOVA did not reveal a significant session effect (*F*(2, 20) = 0.515, *p* = 0.605, *ω^2^* < 0.001), a time effect (*F*(1, 10) = 1.874, *p* = 0.201, *ω^2^* = 0.013, “medium”), or an interaction between time and session (*F*(2, 20) = 0.446, *p* = 0.646, *ω^2^* < 0.001; Figure 3 and Table 3).

### 3.3. Enjoyment

Figure 4 presents the enjoyment scores for women and men during the single and double sessions. The mean enjoyment for women was 107.8 ± 4.6 (85.6 ± 3.7% of the maximum score) and 109.2 ± 5.2 (86.7 ± 4.2% of the maximum score) for the single and double sessions, respectively. A paired Student’s *t*-test showed no statistically significant difference in women’s enjoyment between sessions (Δ = −1.36; 95% CI, −5.32 to 2.60); t(10) = −0.766; *p* = 0.461; d = −0.23; 95% CI, −0.82; 0.37).

For men, the mean enjoyment scores were 104.1 ± 9.7 (82.6 ± 7.7% of the maximum score) and 106.2 ± 7.7 (84.3 ± 6.1% of the maximum score) during the single and double sessions, respectively. No statistically significant difference in men’s enjoyment was found between sessions (Δ = −2.09; 95% CI, −5.41; 1.23; *t*(10) = −1.401; *p* = 0.191; *d* = 0.165; 95% CI, −1.03; 0.20).

### 3.4. Affective Response

The pre-session and post-session affective response scores for women and men are presented in Table 4.

For women, the Friedman test revealed a significant effect on affective response across time points (pre-session and post-session) and interventions (single, double, and control; *X^2^*(5) = 29.065; *p* < 0.001; Kendall’s *W* = 0.528, indicating “moderate agreement”). Post hoc analysis with Bonferroni correction showed that affective responses were significantly lower post-control than post-single (*p* < 0.001), post-double (*p* < 0.001), and pre-control (*p* < 0.001). Moreover, Post hoc analysis with Bonferroni correction demonstrated that affective responses were significantly lower post-double than pre-single (*p* = 0.018) and pre-double (*p* = 0.018).

For men, the Friedman test also showed a significant effect on affective response across time points (pre-session and post-session) and interventions (single, double, and control; *X^2^*(5) = 16.328; *p* = 0.006; Kendall’s *W* = 0.297, indicating “weak agreement”). Post hoc analysis with Bonferroni correction indicated that affective responses were significantly lower post-control than pre-single (*p* = 0.016) and post-double (*p* = 0.028).

### 3.5. Exercise Intensity Elicited by Beach Tennis Sessions

During the single session, the mean heart rate for women was 141.5 (3.5) bpm (77.4% ± 3.1% of HRmax), while during the double session, it was 145.2 ± 2.6 bpm (80.3% ± 4.3% of HRmax). A Wilcoxon test indicated a significantly higher heart rate during the double session than the single session (Δ = −2.99; 95% CI, −5.22 to −0.75; *p* = 0.014; *r_B_* = −0.32; 95% CI, −0.95 to −0.52], “small”; Figure 5 and Table 5).

For men, the heart rate during the single session was 147.3 ± 3.9 bpm (80.4% ± 3.8% of HRmax), while during the double session, it was 143.3 ± 2.6 bpm (77.4% ± 4.5% of HRmax). A paired Student’s *t*-test revealed a significantly higher heart rate during the single session than the double session (Δ = 3.95; *t*(10) = 3.678; *p* = 0.004; *d* = 1.10; 95% CI, 0.33 to 1.85, “small”; Figure 5 and Table 5). Both sessions were classified as vigorous exercise based on %HRmax.

## 4. Discussion

The primary objective of this study was to assess the effect of single and double beach tennis matches on state anxiety symptoms. Additionally, the study aimed to characterize the enjoyment, affective response, and exercise intensity elicited by beach tennis sessions. The study’s key finding was that beach tennis had an anxiolytic effect in women but not in men. Furthermore, the beach tennis sessions elicited strong affective responses and enjoyment in both women and men. Regarding exercise intensity, the sessions resulted in heart rate responses classified as vigorous exercise.

### 4.1. Characterization of the Participants

Statistical analysis revealed higher scores for trait anxiety symptoms in women than in men. The literature consistently shows that women are more prone to develop anxiety disorders, with the prevalence of such disorders being 60% higher in women than in men [4,7,38]. Therefore, our findings align with existing research. This sex-based difference justifies our decision not to include sex as a direct factor in the statistical analysis but rather to conduct separate analyses for women and men. Studies suggest that women, compared to men, display distinct emotional and hormonal characteristics, influencing how they respond to stress-inducing situations [8,39]. Moreover, as noted previously, there is a significant gap in exercise science research on women’s specific physiological responses [32]. Historically, research has predominantly focused on men, contributing to the underrepresentation of female-specific data. By conducting separate analyses, our study aims to address this disparity and provide more inclusive insights relevant to both sports science and mental health. Additionally, the selected age range (18–40 years) was chosen because of the high prevalence of anxiety symptoms, particularly among women in this age group.

### 4.2. State Anxiety Symptoms, Affective Response, and Enjoyment

A session of beach tennis produced an anxiolytic effect in women across both formats but not in men, with state anxiety symptoms after the singles match being statistically lower than after the control session. The observed gender differences in anxiety reduction may be attributed to various factors, including differences in baseline anxiety levels, coping mechanisms, and psychological responses to physical activity. Women may experience greater reductions in anxiety due to their higher baseline levels of state and trait anxiety, which provide more room for improvement. Indeed, in the current study, women reported average pre-intervention scores of approximately 35 (state anxiety level), while men reported scores around 25 (state anxiety level). These values classify women and men as having moderate and mild state anxiety symptoms, respectively. Consequently, the low baseline anxiety scores in men limited the possibility of observing anxiety reduction—a phenomenon referred to as the “floor effect.” Additionally, social interactions during the game might play a more significant role in alleviating anxiety for women, as suggested by studies emphasizing the importance of social support in women’s mental health during physical activity [40]. Indeed, social support plays a crucial role in enhancing women’s mental health, particularly during physical activity. Numerous studies have demonstrated that social relationships fostered through physical activities can significantly improve mental health outcomes for women. For instance, Takeda et al. [41] found that engaging in leisure activities with others is a key factor in preventing mental health deterioration, highlighting the importance of social interactions in exercise contexts. This aligns with findings from Bedaso et al. [42], which indicated that social support is a significant predictor of health-related quality of life, particularly in the mental health domain during pregnancy.

Several studies on non-clinical populations [9,39,43,44,45] have found that physical exercise does not significantly reduce anxiety when baseline levels are already low [43]. To the best of our knowledge, no previous research has examined the effects of a single beach tennis session on anxiety symptoms, making our findings novel. The results suggest that beach tennis could serve as an effective alternative for managing anxiety symptoms, particularly in women.

The inclusion of single and double matches in this study was driven by the need to investigate both modalities commonly practiced in training and competition, providing a more comprehensive understanding of beach tennis as a recreational sport. The findings suggest that both formats can effectively reduce anxiety in women. The anxiolytic effect observed in women may be related to hormonal and psychological differences in stress coping mechanisms, as highlighted in the literature [46]. Furthermore, the social context and recreational environment of beach tennis may have a greater influence on women than on men, as explained above.

Exercise enjoyment plays a key role in adherence to physical activity programs. We evaluated enjoyment using the PACES scale, and participants across both formats and genders reported high enjoyment levels, exceeding 80% of the maximum PACES score. These results are consistent with findings from studies on emerging exercise trends, such as exergames [9,39,43,44,45] and CrossFit [16].

Carpes et al. [17] also documented high enjoyment levels in beach tennis, with participants scoring 117 points (equivalent to 93%) on the PACES scale. In contrast, traditional exercises, such as continuous moderate-intensity treadmill running and reduced-court volleyball, yielded lower enjoyment scores of 60% and 70.9%, respectively [9,47].

Beach tennis, played outdoors, offers an opportunity for physical exercise in natural settings, which has been shown to enhance health outcomes [48,49,50]. Socialization is another key aspect of beach tennis, as it requires at least one other person to play.

We also assessed the participants’ affective responses. Notably, the median affective response remained at least 4.0 points before and after the beach tennis sessions and before the control session. After the control session, however, the median affective response decreased to 3.0 points. This finding suggests that participants experienced elevated affective responses merely by attending the location where the sport was practiced. These results indicate that engaging in beach tennis can positively influence affective responses, promoting self-efficacy and improving attention [49]. Moreover, research suggests that affective responses are closely associated with factors such as adherence to physical exercise, habit formation, and weekly exercise frequency [50]. Therefore, evaluating affective responses during beach tennis sessions provides valuable insights into the emotional impact of the sport. Furthermore, the strong affective responses observed in both single and double matches underscore the sport’s potential to foster psychological well-being and adherence to exercise programs.

### 4.3. Physiological Responses to the Beach Tennis Sessions

According to the American College of Sports Medicine guidelines, the heart rate responses elicited during beach tennis sessions categorize the exercise intensity as vigorous. Thus, it can be inferred that participating in beach tennis for at least 75 min per week would improve cardiorespiratory fitness.

This pattern of heart rate response aligns with findings by Carpes et al. [17], who reported that the average heart rate reserve was 62%, confirming that beach tennis is a vigorous exercise. The study also indicated that participants in recreational sports may experience significant physiological stress while perceiving minimal exertion, likely due to the sport’s focus on enjoyable activities [51]. These findings are consistent with the results of the present study.

As previously mentioned, only eight studies related to beach tennis are available in PubMed, with only Carpes et al. [17] reporting heart rate data. However, their study focused only on the double-playing format and involved participants with hypertension (mean age: 48.4 years). In contrast, the present study recruited healthy young individuals, providing novel reference values for professionals interested in beach tennis.

### 4.4. Limitations, Strengths, and Future Directions

This study has some limitations. First, biochemical markers such as serotonin, endorphin, cortisol, and lactate were not measured, which could have provided further insights into the physiological effects of beach tennis. Second, the study relied on self-reported questionnaires, inventories, and scales, meaning the accuracy of the data depended on participants’ honesty. Third, a limitation of the present study is the small sample size, which may reduce the generalizability of the findings. Additionally, the sample size was not calculated a priori due to the absence of similar studies in the literature to guide this estimation.

Nevertheless, all instruments used were validated for the population evaluated and demonstrated high reproducibility, minimizing potential biases. These limitations do not detract from the study’s conclusions.

The strengths of this study include the following: First, beach tennis, like other forms of physical exercise, is safe, has minimal risk of side effects, and can simultaneously prevent and treat various diseases. Second, approximately 2.15 billion people worldwide—nearly half the global population—live in low-elevation coastal zones [52]. Beach tennis can also be played on sand courts in non-coastal areas, increasing its accessibility.

Future studies should investigate the physiological biomarkers involved in beach tennis and explore the sport’s long-term effects on physical and psychological well-being across diverse populations. To deepen the understanding of anxiety reduction mechanisms, future research could track physical stress markers, such as heart rate variability, salivary cortisol, or catecholamine levels, in addition to self-reported measures of anxiety. These physiological indicators may provide valuable insights into the interplay between physical activity, stress responses, and anxiety regulation. Furthermore, future research could assess other variables, such as anger, and compare beach tennis with other forms of physical exercise.

Additionally, the findings of this study suggest that beach tennis can be a valuable tool for sports-based mental health interventions, particularly for women. The observed anxiolytic effect in women highlights the potential of this sport to manage anxiety symptoms in a practical and engaging manner. Moreover, the strong affective responses and high levels of enjoyment reported by both women and men underscore the appeal of beach tennis as a recreational activity. These attributes may help address common barriers to physical activity, such as lack of motivation and perceptions of exercise as uninteresting.

To leverage these findings, we recommend incorporating beach tennis into existing mental health and wellness initiatives, such as community recreation centers and public health campaigns. Furthermore, future research should explore how structured beach tennis programs could be adapted for diverse populations and settings, ensuring inclusivity and maximizing the sport’s potential as a mental health intervention.

Finally, future studies should aim to include more diverse populations, such as participants from different age groups, cultural backgrounds, and levels of physical activity, to better generalize the findings and explore potential variations in anxiety responses across different demographics.

## 5. Conclusions

Both single and double beach tennis sessions induced anxiolytic effects in healthy women (but not in men), with state anxiety symptoms after the singles match being statistically lower than after the control session. The sessions elicited heart rate responses indicative of vigorous-intensity physical activity for both formats and sexes. Participants also reported high levels of affective responses and enjoyment during the sessions.

This study provides a comprehensive overview of the physiological and psychobiological responses associated with beach tennis. The findings contribute to a deeper understanding of the sport and offer useful insights for training programs and health promotion initiatives. These results are valuable for sports science professionals interested in the potential benefits of beach tennis.

## Figures and Tables

**Figure 1 ijerph-22-00038-f001:**
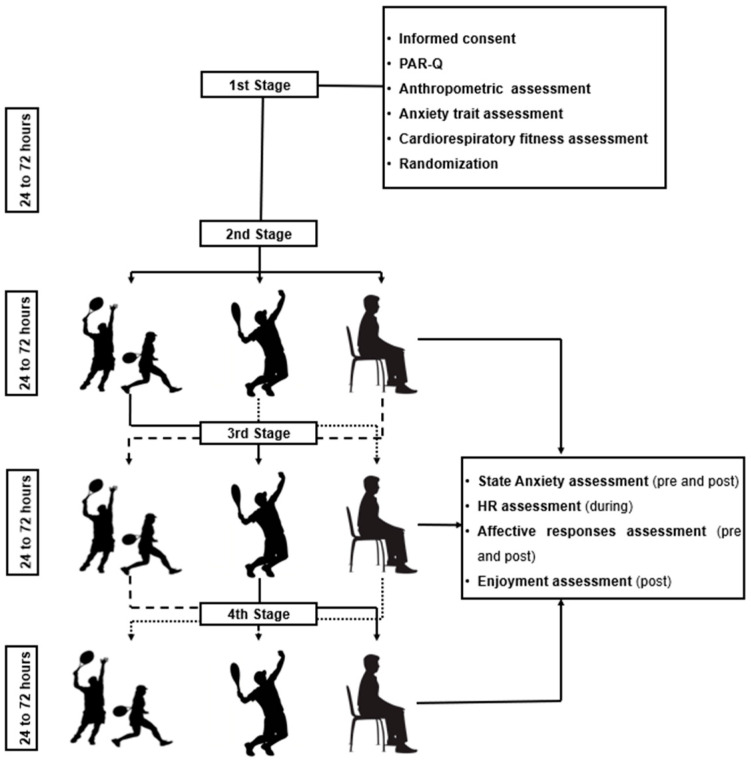
Diagram of the experimental study design. HR, heart rate; PAR-Q, Physical Activity Readiness Questionnaire.

**Figure 2 ijerph-22-00038-f002:**
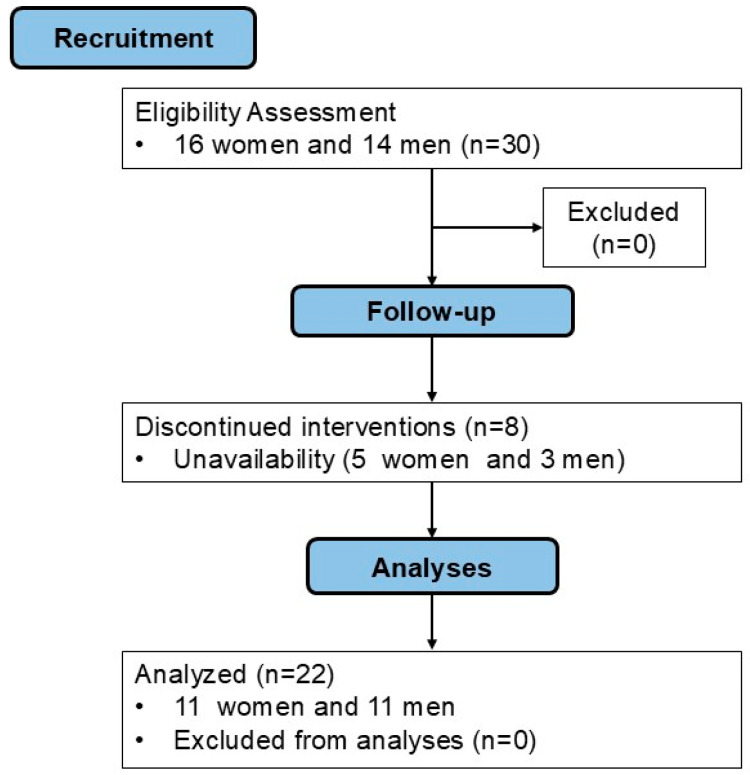
Flow diagram of the study.

**Figure 3 ijerph-22-00038-f003:**
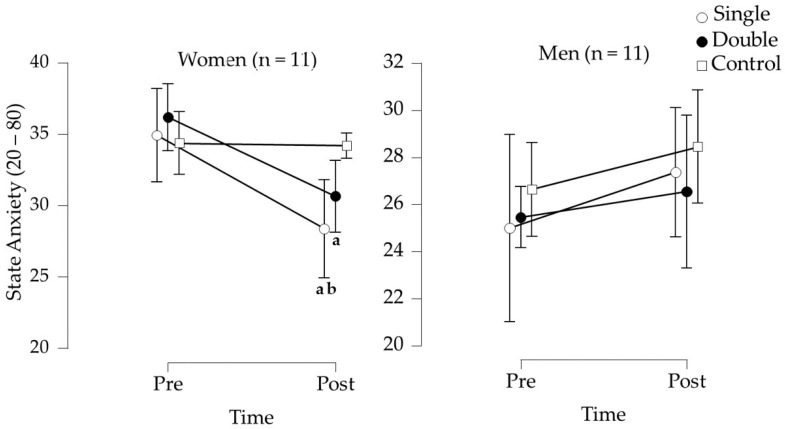
State anxiety symptoms before and after the interventions for women and men beach tennis players. Data are presented as the mean and 95% confidence interval. ^a^ Statistically significant difference between pre- and post-intervention. ^b^ Statistically significant difference compared with the post control session.

**Figure 4 ijerph-22-00038-f004:**
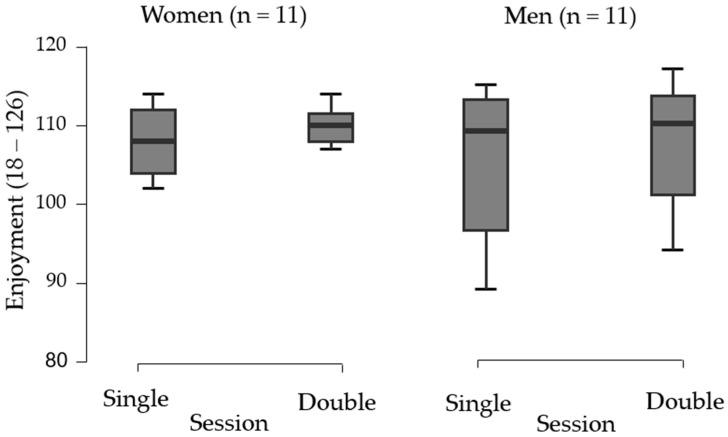
Comparison of enjoyment between single and double sessions for women and men.

**Figure 5 ijerph-22-00038-f005:**
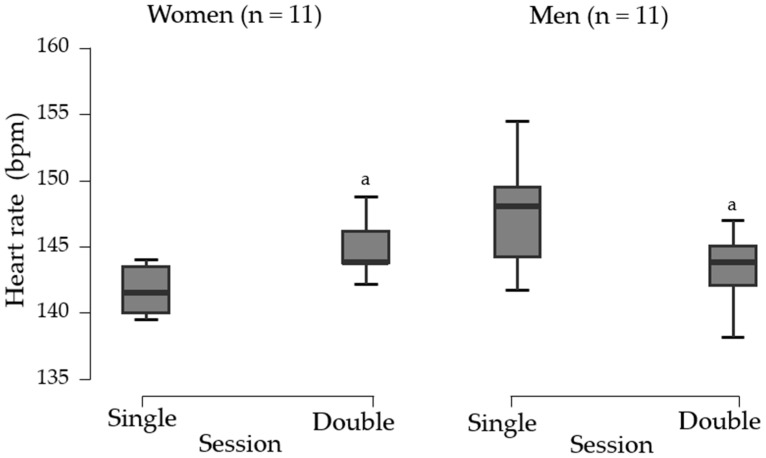
Heart rate comparison between single and double sessions for women. ^a^ Statistically significant difference compared with the single session.

**Table 1 ijerph-22-00038-t001:** Participant characteristics.

	Women (*n* = 11)	Men (*n* = 11)	Overall (*n* = 22)	Δ (95% CI)	Effect Size		*p*-Value
					Value (95%CI)	Classification	
Age (yr)	35.18 ± 5.13	26.00 [14.00] ^a^	35.00 [13.50] ^a^	8.00 (0.00; 14.00) ^b^	−0.47 (−0.76; 0.02) ^c^	medium	0.064 ^d^
Height (m)	1.63 ± 0.04	1.79 ± 0.07	1.71 ± 0.09	−0.15 (−0.20; −0.10) ^e^	−2.65 (1.46; 3.80) ^f^	large	<0.001 ^g^
Body mass (kg)	62.01 ± 5.98	88.08 ± 13.17	69.90 [25.57] ^a^	−26.06 (−35.16; −16.95) ^e^	2.55 (1.38; 3.68) ^f^	large	<0.001 ^g^
Body mass index (kg/m^2^)	23.16 ± 2.67	27.55 ± 4.29	25.36 ± 4.15	−4.39 (−7.57; −1.20) ^e^	1.23 (0.30; 2.13) ^f^	large	0.009 ^g^
Trait anxiety (score)	40.18 ± 8.12	28.00 [5.50] ^a^	35.31 ± 10.43	11.00 (6.00; 18.00) ^b^	−0.61 (−0.84; −0.22) ^c^	large	0.016 ^d^
Experience (months)	17.00 [16.00] ^a^	13.72 ± 5.76	14.50 [14.50] ^a^	2.00 (−5.00; 11.00) ^b^	−0.17 (−0.58; 0.32) ^c^	small	0.526 ^d^
HRmax (bpm)	179.81 ± 6.29	185.54 ± 9.86	181 [7.00] ^a^	−5.72 (−13.08; 1.63) ^e^	0.69 (−0.18; 1.55) ^f^	medium	0.120 ^g^
$\dot{V}O_{2}$max (mL/kg/min)	42.20 [3.60] ^a^	55.12 ± 11.90	44.95 [13.17] ^a^	−11.00 (−17.90; −1.10) ^b^	−0.59 (0.18; 0.82) ^f^	medium	0.019 ^d^

Note: Δ, mean difference; 95% CI, 95% confidence interval; HRmax, maximum heart rate; $\dot{V}O_{2}$max, maximum oxygen uptake. Data are presented as mean ± SD unless otherwise noted. ^a^ Data presented as median and interquartile range. ^b^ Median difference calculated using the Hodges–Lehmann estimate. ^c^ Effect size is given by the rank biserial correlation. ^d^ Mann–Whitney U test, due to data not presenting a normal distribution. ^e^ Mean difference. ^f^ Cohen’s d. ^g^ Student’s *t*-test.

**Table 2 ijerph-22-00038-t002:** Participants’ experience with beach tennis.

Variables	Women (*n* = 11)	Men (*n* = 11)	Overall (*n* = 22)
Training frequency (%)			
2 times/week	4 (36.4)	6 (54.5)	10 (45.5)
3 times/week	3 (27.3)	2 (18.2)	5 (22.7)
4 times/week	1 (9.1)	1 (9.1)	2 (9.1)
5 times/week	3 (27.3)	2 (18.2)	5 (22.7)
Purpose (%)			
Recreation	6 (54.5)	11 (100)	17 (77.3)
Competition	5 (45.5)	0 (0.0)	5 (22.7)
Has participated in competitions? (%)			
Yes	11 (100)	8 (72.7)	19 (86.4)
No	0 (0.0)	3 (27.3)	3 (13.6)
How many competitions? (%)			
0	0 (0.0)	3 (27.3)	3 (13.6)
1	3 (27.3)	2 (18.2)	5 (22.7)
2	0 (0.0)	0 (0.0)	0 (0.0)
3	3 (27.3)	1 (9.1)	4 (18.2)
4	1 (9.1)	2 (18.2)	3 (13.6)
5	1 (9.1)	2 (18.2)	3 (13.6)
6	0 (0.0)	1 (9.1)	1 (4.5)
7	3 (27.3)	0 (0.0)	3 (13.6)
Injury in beach tennis? (%)			
Yes	0 (0.0)	0 (0.0)	0 (0.0)
No	11 (100)	11 (100)	22 (100)
Practices another physical exercise? (%)			
Yes	6 (54.5)	9 (81.8)	15 (68.2)
No	5 (45.5)	2 (18.2)	7 (31.8)
What physical exercise? (%)			
None	5 (45.5)	1 (9.1)	6 (27.3)
Strength training	4 (36.4)	10 (90.9)	14 (63.6)
Crossfit	0 (0.0)	0 (0.0)	0 (0.0)
Footvolley	1 (9.1)	0 (0.0)	1 (4.5)
Tennis	1 (9.1)	0 (0.0)	1 (4.5)

**Table 3 ijerph-22-00038-t003:** Results from a repeated measures analysis of variance (ANOVA) analyzing the interaction between conditions (single session × double session × control session) and time (pre-game × post-game) on participants’ state anxiety levels.

Sex	Main Factors				Post Hoc Comparisons			Mean Difference (95% CI) ^c^	t	*p*-Value	Effect Size	
	Factors	F	*p*-Value	ω^2^_(classification)_	Session	Pre	Post				d (95% CI)	Classification
Women (n = 11)	Session	3.377	0.054	0.019_(small)_	Single	34.90 ± 10.05	28.36 ± 3.90 ^a,b^	−0.005 (−0.009; 0.001)	−4.036	0.007	−0.744 (−1.615; −0.104)	medium
	Time	11.706	0.007	0.054_(small)_	Double	36.18 ± 8.95	30.63 ± 7.31 ^a^	−0.005 (−0.009; 8.326 × 10^−4^)	−3.903	0.010	−0.720 (−1.578; −0.139)	medium
	Session × Time	9.059	0.002	0.034_(small)_	Control	34.36 ± 9.40	34.18 ± 7.54	3.555 × 10^−4^ (−0.004; 0.005)	0.278	1.000	0.051 (−0.596; 0.699)	trivial
Men (n = 11)	Session	0.515	0.605	<0.001_(trivial)_	Single	25.00 ± 5.25	27.36 ± 5.53	0.004 (−0.003; 0.011)	1.642	1.000	0.456 (−0.502; 1.414)	small
	Time	1.874	0.201	0.013_(small)_	Double	25.45 ± 4.92	26.54 ± 6.39	0.001 (−0.006; 0.008)	0.632	1.000	0.175 (−0.733; 1.083)	trivial
	Session × Time	0.446	0.646	<0.001_(trivial)_	Control	26.63 ± 5.73	28.45 ± 7.47	0.002 (−0.005; 0.009)	0.841	1.000	0.233 (−0.682; 1.148)	small

Note: Data are presented as mean ± standard deviation. CI: confidence interval. ω^2^*:* omega squared effect size. d: Cohen’s d effect size. ^a^ Statistically significant difference between pre- and post-intervention. ^b^ Statistically significant difference between post-control and post-single session (Mean difference (95% CI) = 0.005 (0.001; 0.009); t = 4.278; *p* = 0.002; d (95% CI) = 0.777 (−0.104; 1.657) (medium). ^c^: Mean difference (95% CI) calculated from inverse-transformed values due to the non-normal distribution of state anxiety.

**Table 4 ijerph-22-00038-t004:** Affective response scores for women (n = 11) and men (n = 11) across sessions.

Sex	Main Factors				Post Hoc Comparisons			t	*p*-Value	Effect Size	
	Factors	X^2^	*p*-Value	Kendall’s *W* _(classification)_	Session	Pre	Post			r_B_ (95% CI)	Classification
Women	Session	29.065	<0.001	0.528 (moderate)	Single	4.0 [2.0] ^b^	5.0 [1.0] ^a^	2.626	0.172	−1.000	large
					Double	4.0 [0.5] ^b^	5.0 [0.0] ^a^	3.440	0.018	−1.000	large
					Control	5.0 [0.5] ^a^	3.0 [1.5]	5.704	<0.001	1.000	large
Men	Session	16.328	0.006	0.297 (weak)	Single	5.0 [0.0] ^a^	5.0 [1.5] ^a^	0.870	0.388	0.524	large
					Double	5.0 [1.5]	5.0 [0.5]	0.469	1.000	−0.267	small
					Control	4.0 [1.0] ^a^	3.0 [1.5]	2.276	0.407	1.000	large

Note: Data are presented as median [interquartile range]. ^a^ Statistically significant difference compared with the post-control session. ^b^ Statistically significant difference compared with the post-double session. r_B_: rank-biserial correlation coefficient.

**Table 5 ijerph-22-00038-t005:** Heart rate scores for women (n = 11) and men (n = 11) across sessions.

Sex	Session	Mean ± SD	Mean Difference	*p*-Value	Effect Size	Classification
			(95% CI)		d (95% CI)	
Women	Single	141.5 [3.5] ^b^	−2.995 (−5.225; −0.750) ^c^	0.014	0.328 (−0.958; −0.525) ^d^	small
	Double	145.2 ± 2.6 ^a^				
Men	Single	147.3 ± 3.9	3.950 (1.557; 6.343) ^e^	0.004	1.109 (0.332; 1.853) ^f^	small
	Double	143.3 ± 2.6 ^a^				

Note: 95% CI, 95% confidence interval; Data are presented as mean ± SD unless otherwise noted. ^a^ Statistically significant difference compared with the single session ^b^ Data presented as median and interquartile range. ^c^ Median difference calculated using the Hodges–Lehmann estimate. ^d^ Effect size is given by the rank biserial correlation. ^e^ Mean difference. ^f^ Cohen’s d.

## Data Availability

The data that support the findings of this study are available from the corresponding author upon reasonable request.

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
