# Peer review of "A Single Session of Beach Tennis with Recreational Athletes Improves Anxiety Symptoms in Women but Not in Men: A Randomized Trial"

_ijerph, 2024, doi:10.3390/ijerph22010038_

Round 1
Reviewer 1 Report
Comments and Suggestions for Authors
· The percentage of plagiarism in the manuscript has to be reduced
· The title is too lengthy and confusing. The average number of words in the title should be between 13-16
· What is the reason for quoting specific data about Brazil?Is this a regional-specific study?
· The introduction is poorly written. There are a number of sentences written without any coherence, and there is no logical flow. The third paragraph is just a repetition of the first paragraph
· What is the rational of conducting this study .It is not properly presented
· The literature review in the introduction lacks depth.
· The objective of the study is not in line with the title
· The methodology section is poorly written and does not follow the general format. Line 96- 98 should be in the results
· Eligibility criteria is incomplete
· Was the sample size calculated prior to the data collection
· The participants in the study are recreational athletes who train from 2- 5 times per week. Did they instructed to stop regular training during the study period?
· The design is not clear. How many times the measurement was taken? What is the rationale for performing single and double sessions separately? What difference the authors were expecting?
· What is the reliability and validity of the tools used in data collection
· Please provide the rationale for using these outcome measures
· Figure 4 and figure 5 should be replaced by tables for clarity .
· Discussion lacks depth, and there is no critical analysis of the results.
Author Response
Manuscript number: ijerph-3318686
Title: A Single Session of Beach Tennis Improves Anxiety Symptoms in Women But Not in Men: A Randomized Trial (former title: A Single Session of Beach Tennis Improves Anxious Symptoms in Healthy Women But Not in Healthy Men Recreational Players: A Within-Trial Randomized Clinical Trial)
Mia Sevic
Assistant Editor
International Journal of Environmental Research and Public Health
Dear Mia Sevic,
Thank you for the opportunity to resubmit our revised manuscript. We have carefully addressed all the reviewers’ comments and incorporated the requested revisions, clearly marked using MS Word’s track changes feature. Each reviewer’s question has been thoroughly addressed, and we believe these changes have significantly enhanced the quality of the manuscript. We are confident that the revised version aligns with the high publication standards of International Journal of Environmental Research and Public Health.
- The percentage of plagiarism in the manuscript has to be reduced
Response: Thank you for your feedback. We understand the importance of ensuring originality in our manuscript and take such concerns very seriously.
To address your comment, we ran the manuscript through plagiarism detection software (CopySpider), and no significant level of similarity was found (see attached the report). We are committed to maintaining the highest ethical standards and are willing to make any necessary adjustments.
Could you kindly indicate specific sections or sentences where you believe the overlap occurs? This will help us address your concerns more effectively and ensure the quality of our submission.
We appreciate your guidance and look forward to your response.
- The title is too lengthy and confusing. The average number of words in the title should be between 13-16
Response: Thank you for your valuable feedback regarding the title of our manuscript. We understand the importance of having a concise and clear title to accurately represent the study's content. Based on your suggestion, we have revised the title to make it shorter and more focused, while still conveying the key elements of the study:
Revised Title:
"A Single Session of Beach Tennis Improves Anxiety Symptoms in Women but Not in Men: A Randomized Trial"
We believe this revised version aligns better with the recommended word count and retains the essence of the study. Please let us know if further adjustments are needed.
Thank you for your constructive input.
- What is the reason for quoting specific data about Brazil?Is this a regional-specific study
Response: Thank you for your question regarding the inclusion of specific data about Brazil.
The primary reason for referencing Brazil is that the data collection for this study was conducted in the country. Furthermore, Brazil is known to have the highest prevalence of anxiety disorders globally, as reported by the World Health Organization. This context underscores the importance of developing and studying effective strategies for managing anxiety symptoms in this population.
We believe this information is relevant to emphasize the necessity and potential impact of research aimed at addressing anxiety, particularly in a country where it poses a significant public health challenge.
We hope this clarifies the rationale behind including data about Brazil. Please let us know if further elaboration is needed.
- The introduction is poorly written. There are a number of sentences written without any coherence, and there is no logical flow. The third paragraph is just a repetition of the first paragraph
Response: Thank you for your feedback regarding the introduction. We have carefully reviewed your comments and taken significant steps to address the concerns raised. Specifically, we have:
- Deleted the third paragraph, as it was indeed repetitive.
- Conducted an extensive revision of the entire introduction, making substantial changes to improve coherence, logical flow, and overall clarity.
We believe these adjustments have enhanced the readability and structure of the introduction. Please let us know if further refinements are needed.
- What is the rational of conducting this study .It is not properly presented
Response: Thank you for your feedback regarding the rationale for conducting this study. We would like to respectfully point out that the rationale is explicitly presented in the penultimate paragraph of the introduction:
"Considering the rising number of beach tennis practitioners, the limited research on this sport, the high prevalence of anxiety disorders (particularly among women), and the role of motivation in physical activity adherence, a comprehensive investigation is warranted."
We believe this clearly outlines the motivation and relevance of the study. Furthermore, we note that this aspect was not raised as a concern by any of the other reviewers, which suggests that this explanation has been generally well-received.
Nonetheless, if you believe further clarification is necessary, we are willing to refine this section to make the rationale even more explicit. We remain committed to ensuring the manuscript meets the highest standards of clarity and scientific rigor.
Thank you for your attention and for your efforts in improving our work.
- The literature review in the introduction lacks depth.
Response: Thank you for your observation regarding the depth of the literature review in the introduction. As mentioned in our response to Comment 4, the introduction has been extensively revised, including substantial updates to enhance its coherence, logical flow, and depth.
We believe these changes have addressed the concern and significantly improved the quality of the literature review. Please let us know if there are additional aspects that require further attention.
- The objective of the study is not in line with the title
Response: Thank you for pointing out a potential misalignment between the study's objective and the title. We have carefully reviewed both elements and adjusted the study objective to ensure consistency with the title. The revised objective now reads:
"This study aimed to evaluate and compare the acute effects of a beach tennis session on anxiety symptoms in women and men. Secondary objectives included characterizing enjoyment, affective responses, and exercise intensity (measured by heart rate) during the session."
We believe this revised version better aligns with the title, emphasizing the comparison of anxiety symptom changes between women and men while maintaining the additional aspects of the study.
We appreciate your input and remain open to further feedback to refine the manuscript.
- The methodology section is poorly written and does not follow the general format. Line 96- 98 should be in the results
Response: Thank you for your feedback regarding the methodology section and the placement of specific content. To address your concerns, we have:
- Moved Tables 1 and 2, as well as Figure 1 (now renumbered as Figure 2), to the Results section, as suggested.
- Made significant revisions to the Methods section to improve its structure, clarity, and adherence to the general format.
We believe these changes have enhanced the overall organization and readability of the manuscript. Please let us know if there are additional adjustments needed.
- Eligibility criteria is incomplete
Response: Thank you for your feedback. While we consider your suggestion regarding the eligibility criteria somewhat generic, we have nonetheless made significant revisions to provide more detailed information. The revised section now clearly outlines the inclusion and exclusion criteria, along with the rationale for each one.
- Was the sample size calculated prior to the data collection
Response: An “a priori” power calculation was not performed because no similar studies were found in the literature. On the other hand, the “post hoc” power calculations have been recently criticized in recent literature (see DOI:10.1002/gepi.22464). For these reasons, we included the effect sizes and the confidence intervals for a better interpretation and understanding of the true impact of the current results.
- The participants in the study are recreational athletes who train from 2- 5 times per week. Did they instructed to stop regular training during the study period?
Response: Thank you for your valuable observation regarding participants' training routines during the study period. To address your concern, we have clarified this aspect in the revised manuscript. Specifically, we have added the following details:
"Participants were requested to wear appropriate gear for physical activity. Stimulants, such as caffeine or tea, were avoided for at least 8 h before each visit. Furthermore, participants were instructed to avoid eating within 2 h before each visit and to follow their regular hydration routine. No training sessions were allowed on the day of laboratory visits, and participants were instructed to refrain from intense physical exercise for at least 24 h prior to each visit."
We hope this addition addresses your concern and provides the necessary clarity about the instructions given to participants to minimize potential confounding factors.
Thank you again for your insightful feedback, which has helped us improve the manuscript. Please do not hesitate to let us know if there are further details you would like us to address.
- The design is not clear. How many times the measurement was taken? What is the rationale for performing single and double sessions separately? What difference the authors were expecting?
Response: Thank you for raising this point. We believe the number and timing of the measurements were already clear in the original submission, as stated in the manuscript:
"State anxiety symptoms and affective responses were measured before and after each intervention. Enjoyment was assessed only after the beach tennis sessions, and heart rate was recorded throughout all interventions."
To further clarify, we have now included the specification that state anxiety symptoms and affective responses were measured immediately before and after, and enjoyment was assessed immediately after the beach tennis sessions.
Regarding the rationale for evaluating single and double matches separately, these two modalities are integral to beach tennis training and competitions. As such, understanding their distinct impacts on psychological and physiological outcomes is necessary for advancing knowledge in this area.
We hope these clarifications address your concerns. Thank you for your constructive feedback, which has helped us refine the manuscript.
- What is the reliability and validity of the tools used in data collection
Response: Thank you for your question regarding the reliability and validity of the tools used in data collection. To address this, we have included Cronbach's alpha values or Intraclass Correlation Coefficients (ICC) for the instruments employed in the study. These details are now specified in the methodology section of the revised manuscript.
- Please provide the rationale for using these outcome measures
Response: Thank you for your query regarding the rationale behind our choice of outcome measures. The instruments used in this study were selected because they are:
- Validated: These measures have demonstrated strong reliability and validity in previous studies.
- Cost-effective: Their implementation does not require expensive equipment, ensuring accessibility.
- Culturally Adapted: The instruments have been translated and culturally adapted to Brazilian Portuguese, allowing for accurate assessment within our study population.
We believe these points highlight the appropriateness of our chosen measures for addressing the study objectives.
We have included the following sentence in manuscript to clarify:
“The outcome measures employed in this study were selected due to their robust validation, cost-effectiveness, and prior cultural adaptation to Brazilian Portuguese, ensuring reliable and accurate assessment within the studied population.”
- Figure 4 and figure 5 should be replaced by tables for clarity.
Response: Thank you for your suggestion to replace Figures 4 and 5 with tables for improved clarity. In response, we have included corresponding tables to present the data in a detailed and structured format. However, we opted to retain the figures as well, as they provide a visual summary that enhances the reader’s ability to quickly grasp key patterns and trends in the data.
We believe that presenting both tables and figures offers a complementary approach, catering to different preferences for data interpretation. We hope this solution meets your expectations and provides greater clarity to the presentation of our results.
- Discussion lacks depth, and there is no critical analysis of the results.
Response: We would like to thank you for your comments. While we believe the suggestions provided were somewhat generic, we have made the necessary adjustments to improve the clarity and depth of the discussion, as outlined below:
- Justification for Separate Analysis by Sex
We added more details to explain why we conducted separate analyses for men and women. Specifically, we discussed the differences in baseline anxiety symptoms between sexes and the role of hormonal and psychological factors in stress responses. We also addressed the lack of female-specific data in exercise science research, emphasizing the importance of considering gender differences. - Influence of Social Context and Recreational Environment on Women
We elaborated on how the recreational and social environment of beach tennis may have a stronger influence on women than men, referencing literature that highlights psychological and social factors in stress coping mechanisms. - Clarification of the Use of Single and Double Match Formats
We clarified that the inclusion of both single and double match formats was intentional to capture widely practiced game modalities in beach tennis, with the aim of providing a more comprehensive understanding of the sport. - In-depth Discussion on Affective Responses and Adherence Potential
We expanded the discussion on affective responses, emphasizing their connection to the potential for beach tennis to enhance adherence to physical activity and psychological well-being. - Clarification of Validated Instruments and Contextual Impact
We reinforced the rationale behind using validated instruments in Portuguese, specifically adapted for the Brazilian context, to ensure the relevance and accuracy of the data. - Suggestions for Future Research
We suggested directions for future research, particularly in the inclusion of physiological biomarkers and comparison with other forms of exercise.

Reviewer 2 Report
Comments and Suggestions for Authors
Reviewer Comments
This study offers valuable insights into how beach tennis can reduce anxiety in women, which is a timely and relevant topic. However, a few areas could be improved:
- Introduction:
- Adding context about gender differences in anxiety responses would enhance the introduction. Including references to mental health benefits from other racket sports could also strengthen the rationale.
- Methods:
- The small sample size (22 participants) is a limitation. Clarifying how "healthy" participants were selected and if they were screened for anxiety levels would improve the design.
- Results:
- A chart or table showing pre- and post-intervention anxiety scores would help visualize gender differences more clearly.
- Discussion:
- It would be insightful to explore why women experienced anxiety reduction while men did not. Discussing practical applications of these findings, like promoting beach tennis for stress relief, would add relevance.
- Conclusion:
- Emphasizing the need for more research with diverse samples and suggesting methods to track physical stress markers could provide new directions for future studies.
Overall, the study is interesting and well-conducted, but with minor revisions, it can have an even greater impact. It was recommended for publication with minor revisions.
Comments on the Quality of English LanguageThe manuscript is generally well-written, with clear communication of ideas. However, a few improvements could enhance clarity:
- Clarity and Conciseness: Some sentences are long and could be simplified for better readability.
- Technical Terms: While technical language is appropriate, clearer explanations can help reach a broader audience.
- Grammar and Syntax: There are minor grammatical errors and awkward phrases that need proofreading.
- Transitions: Smoother transitions between sections would improve overall coherence.
Addressing these points will significantly improve the quality of the manuscript's English language.
Author Response
Manuscript number: ijerph-3318686
Title: A Single Session of Beach Tennis Improves Anxiety Symptoms in Women But Not in Men: A Randomized Trial (former title: A Single Session of Beach Tennis Improves Anxious Symptoms in Healthy Women But Not in Healthy Men Recreational Players: A Within-Trial Randomized Clinical Trial)
Mia Sevic
Assistant Editor
International Journal of Environmental Research and Public Health
Dear Mia Sevic,
Thank you for the opportunity to resubmit our revised manuscript. We have carefully addressed all the reviewers’ comments and incorporated the requested revisions, clearly marked using MS Word’s track changes feature. Each reviewer’s question has been thoroughly addressed, and we believe these changes have significantly enhanced the quality of the manuscript. We are confident that the revised version aligns with the high publication standards of International Journal of Environmental Research and Public Health.
Response to Reviewer 2 Comments
- Adding context about gender differences in anxiety responses would enhance the introduction. Including references to mental health benefits from other racket sports could also strengthen the rationale.
Response: Thank you for your insightful comments and suggestions. We have addressed your request to enhance the introduction by incorporating context regarding gender differences in anxiety responses and the potential mental health benefits of other racket sports.
To address the first point, we added the following text to the introduction:
"Brazil has the highest global prevalence of anxiety disorders, with 9.3% of the population affected. Young adults, in particular, experience higher rates of anxiety, with women disproportionately affected compared to men due to a combination of biological, social, and cultural factors. Research indicates that women report higher levels of both state and trait anxiety, potentially influenced by physiological differences, such as hormonal fluctuations associated with the menstrual cycle, which may exacerbate anxiety symptoms. These findings underscore the importance of exploring and developing tailored treatments for anxiety, with a particular focus on addressing the unique needs of women."
This addition emphasizes the importance of considering gender-specific factors in anxiety research, aligning with the study's focus on exploring symptoms in women.
Regarding the second point, we included the following paragraph in the discussion:
"With regard to racket sports and anxiety symptoms, Conde-Ripoll et al. analyzed pre- and post-competitive anxiety and self-confidence in relation to match outcomes among 11 high-level male padel players from Finland. The study found that losing players experienced significant increases in cognitive, somatic, and state anxiety, along with a notable decline in self-confidence, while winning players showed only a significant increase in state anxiety. The authors concluded that players should develop mental skills to manage errors and defeats, and coaches are encouraged to include pressure training and foster a winning mentality during simulated matches. However, this study did not include female athletes."
This text highlights the existing research on racket sports and anxiety while addressing the gap in studies involving women.
We hope these revisions address your concerns and enhance the clarity and relevance of our manuscript. Thank you for helping us strengthen our work.
- The small sample size (22 participants) is a limitation. Clarifying how "healthy" participants were selected and if they were screened for anxiety levels would improve the design.
Response:
Thank you for your comments regarding the sample size and participant selection process. We have addressed your concerns as follows:
- Sample Size:
We recognize that the small sample size (22 participants) is a limitation of our study. To address this, we added the following statement to the "Limitations" section:
"A limitation of the present study is the small sample size, which may reduce the generalizability of the findings. Additionally, the sample size was not calculated a priori due to the absence of similar studies in the literature to guide this estimation."
- Participant Selection and Anxiety Screening:
The participants were classified as "healthy" based on their responses to the Physical Activity Readiness Questionnaire (PAR-Q), ensuring they were free of contraindications to physical activity. Furthermore, to characterize the participants' anxiety levels, we included the results of their trait anxiety scores in Table 1, providing a clear overview of their baseline anxiety levels.
We hope these clarifications and updates adequately address your concerns. Thank you for helping us improve the rigor and transparency of our manuscript.
- A chart or table showing pre- and post-intervention anxiety scores would help visualize gender differences more clearly.
Response:
Thank you for your suggestion. In response to Reviewer #1's request, we have replaced the figure previously used to present these data with a table. This change allows for a clearer and more detailed visualization of the anxiety scores before and after the intervention, stratified by gender.
We hope this modification meets your expectations and improves the clarity of our results. Thank you for your valuable feedback.
- It would be insightful to explore why women experienced anxiety reduction while men did not. Discussing practical applications of these findings, like promoting beach tennis for stress relief, would add relevance.
Response:
Thank you for highlighting the need to explore gender differences in anxiety reduction and suggesting practical applications of our findings. We have addressed your comments as follows:
- Exploring Gender Differences in Anxiety Reduction:
We added the following discussion to explore possible explanations for why women experienced anxiety reduction while men did not:
"The observed gender differences in anxiety reduction may be attributed to various factors, including differences in baseline anxiety levels, coping mechanisms, and psychological responses to physical activity. Women may experience greater reductions in anxiety due to their higher baseline levels of state and trait anxiety, which provide more room for improvement. Additionally, social interactions during the game might play a more significant role in alleviating anxiety for women, as suggested by studies emphasizing the importance of social support in women's mental health during physical activity"
- Practical Applications:
To address the practical implications, we included the following statement in the conclusion:
" Furthermore, the findings of this study suggest that beach tennis can be a valuable tool for sports-based mental health interventions, particularly for women. The observed anxiolytic effect in women highlights the potential of this sport to manage anxiety symptoms in a practical and engaging manner. Furthermore, the strong affective responses and high levels of enjoyment reported by both women and men underscore the appeal of beach tennis as a recreational activity. These attributes may help address common barriers to physical activity, such as lack of motivation and perceptions of exercise as uninteresting.
To leverage these findings, we recommend incorporating beach tennis into existing mental health and wellness initiatives, such as community recreation centers and public health campaigns. Moreover, future research should explore how structured beach tennis programs could be adapted for diverse populations and settings, ensuring inclusivity and maximizing the sport's potential as a mental health intervention."
We hope these additions enhance the depth and applicability of the manuscript. Thank you for your valuable suggestions.
- Emphasizing the need for more research with diverse samples and suggesting methods to track physical stress markers could provide new directions for future studies.
Response:
Thank you for your insightful comment regarding the importance of diverse samples and tracking physical stress markers in future studies. We have incorporated the following points into the "Limitations and Future Directions" section to address your suggestions:
- Diverse Samples:
We added the following statement to emphasize the need for broader research:
"Future studies should aim to include more diverse populations, such as participants from different age groups, cultural backgrounds, and levels of physical activity, to better generalize the findings and explore potential variations in anxiety responses across different demographics."
- Tracking Physical Stress Markers:
We also suggested incorporating physiological markers in future research:
"To deepen the understanding of anxiety reduction mechanisms, future research could track physical stress markers, such as heart rate variability, salivary cortisol, or catecholamine levels, in addition to self-reported measures of anxiety. These physiological indicators may provide valuable insights into the interplay between physical activity, stress responses, and anxiety regulation."
We hope these additions provide meaningful directions for future research and address your suggestions thoroughly. Thank you for helping us improve the relevance and scope of our manuscript.
Reviewer 3 Report
Comments and Suggestions for Authors
The introduction effectively frames the relevance of beach tennis as a mental health intervention, especially for managing anxiety. However, including additional references that explore the relationship between various forms of exercise and anxiety would enhance the background context.
Consider briefly comparing the mental health benefits of beach tennis to those of other popular recreational activities to give readers a broader perspective on this sport's potential advantages.
The randomized crossover clinical trial design is well-suited to your study’s objectives. Detailed descriptions of the participant selection criteria, measurement tools, and session protocols add rigor and transparency to your methodology.
To strengthen the study’s scientific foundation, future research could incorporate physiological measures (e.g., cortisol levels or heart rate variability) to support self-reported anxiety changes with objective data.
Your results are presented clearly, with statistical analysis details that aid in understanding the effects observed in both women and men. The distinct presentation for gender-based results enhances readability.
Adding more visual aids (such as graphs or summarized tables) could facilitate quick reference to key findings, particularly in the results section.
The discussion provides a thoughtful interpretation of the findings, highlighting the observed anxiolytic effect in women and exploring the possible "floor effect" in men’s anxiety scores. Additionally, the acknowledgment of limitations and areas for future research strengthens the credibility of your conclusions.
Expanding on the potential applications of these findings in sports-based mental health interventions would underscore the practical implications of your work. This could include recommendations for incorporating beach tennis into community-based mental health programs.
The language quality is high overall, and the study is well-organized and easy to follow. Some complex sentences could be simplified for enhanced readability, especially in the methods and results sections.
Consider minor adjustments in sentence structure for complex or multi-clause sentences, particularly in sections with detailed statistical information, to improve readability.
Your study addresses a novel topic with rigorous methods, yielding valuable insights for sports science and mental health. With minor adjustments, such as additional visual data and expanded discussion on broader applications, your work could make an even stronger impact in the field. Thank you for your contribution.
Comments on the Quality of English LanguageThe quality of the English language in this manuscript is high, with clear and precise explanations throughout. The scientific terminology is appropriately used, and the language effectively communicates the study's purpose, methodology, and findings.
A few sentences, particularly in the methods and results sections, are complex or contain multiple clauses. Simplifying these sentences or breaking them into shorter ones could improve readability, especially for non-expert readers.
Example: Instead of “The study’s key finding was that beach tennis had an anxiolytic effect in women but not men,” consider restructuring to: “The key finding of this study is the anxiolytic effect of beach tennis observed in women, but not in men.”
The flow between sections is generally smooth. However, some technical details in the methods section might benefit from simpler language or clarifications, especially when describing statistical tests and measurement tools.
Using consistent terminology, such as “single session” and “double session,” throughout the paper helps, but ensure that any abbreviations or terms introduced are defined the first time they appear.
The English language quality is very good, with minor adjustments recommended for enhanced clarity and flow.
Author Response
Manuscript number: ijerph-3318686
Title: A Single Session of Beach Tennis Improves Anxiety Symptoms in Women But Not in Men: A Randomized Trial (former title: A Single Session of Beach Tennis Improves Anxious Symptoms in Healthy Women But Not in Healthy Men Recreational Players: A Within-Trial Randomized Clinical Trial)
Mia Sevic
Assistant Editor
International Journal of Environmental Research and Public Health
Dear Mia Sevic,
Thank you for the opportunity to resubmit our revised manuscript. We have carefully addressed all the reviewers’ comments and incorporated the requested revisions, clearly marked using MS Word’s track changes feature. Each reviewer’s question has been thoroughly addressed, and we believe these changes have significantly enhanced the quality of the manuscript. We are confident that the revised version aligns with the high publication standards of International Journal of Environmental Research and Public Health.
Response to Reviewer 3 Comments
- The introduction effectively frames the relevance of beach tennis as a mental health intervention, especially for managing anxiety. However, including additional references that explore the relationship between various forms of exercise and anxiety would enhance the background context.
Response: Thank you for your insightful suggestion. We have included additional references that explore the relationship between various forms of exercise and anxiety to enhance the background context in the introduction, as requested.
- Consider briefly comparing the mental health benefits of beach tennis to those of other popular recreational activities to give readers a broader perspective on this sport's potential advantages.
Response: Thank you for your suggestion. We may not have fully understood your comment. To the best of our knowledge, no studies have investigated the effects of beach tennis on mental health to date. If this does not address your doubt, please let us know, and we will be happy to clarify further.
- The randomized crossover clinical trial design is well-suited to your study’s objectives. Detailed descriptions of the participant selection criteria, measurement tools, and session protocols add rigor and transparency to your methodology.
Response: Thank you for your positive feedback on the study design and methodology. We appreciate your acknowledgment of the randomized crossover clinical trial approach.
- To strengthen the study’s scientific foundation, future research could incorporate physiological measures (e.g., cortisol levels or heart rate variability) to support self-reported anxiety changes with objective data.
Response: Thank you for your suggestion to incorporate physiological measures in future research to strengthen the study’s scientific foundation. We have included this information in the "Limitations " section to address this point.
- Your results are presented clearly, with statistical analysis details that aid in understanding the effects observed in both women and men. The distinct presentation for gender-based results enhances readability.
Response: Thank you for your insightful suggestion. We completely agree with your point, and we had already highlighted the absence of physiological measures as a limitation of our study. Additionally, in initial version of manuscript, we included this aspect as a perspective for future research to complement self-reported anxiety changes with objective data.
- Adding more visual aids (such as graphs or summarized tables) could facilitate quick reference to key findings, particularly in the results section.
Response: Thank you for your suggestion. The manuscript currently includes five figures and three tables to illustrate the key findings and facilitate understanding. If you have a specific recommendation for additional visual aids or improvements to the existing ones, we would greatly appreciate your guidance to enhance the clarity and impact of the results section.
- The discussion provides a thoughtful interpretation of the findings, highlighting the observed anxiolytic effect in women and exploring the possible "floor effect" in men’s anxiety scores. Additionally, the acknowledgment of limitations and areas for future research strengthens the credibility of your conclusions.
Response: Thank you for your encouraging feedback. We are glad that you found the discussion to provide a thoughtful interpretation of the findings, as well as a balanced acknowledgment of limitations and future research directions. Your comments reinforce our effort to ensure a comprehensive and credible discussion.
- Expanding on the potential applications of these findings in sports-based mental health interventions would underscore the practical implications of your work. This could include recommendations for incorporating beach tennis into community-based mental health programs.
Response: Thank you for your suggestion. We have expanded the discussion to include the potential applications of our findings in sports-based mental health interventions. Specifically, we highlighted how beach tennis could be incorporated into community-based mental health programs, emphasizing its anxiolytic effects in women, the strong affective responses and enjoyment elicited in both sexes, and its vigorous exercise intensity. We also proposed practical recommendations for integrating beach tennis into public health and wellness initiatives while addressing common barriers to physical activity. These additions underscore the practical implications of our work, as suggested.
- The language quality is high overall, and the study is well-organized and easy to follow. Some complex sentences could be simplified for enhanced readability, especially in the methods and results sections.
Response: Thank you for your feedback regarding the language quality and readability of the manuscript. We would like to inform you that the article was initially reviewed by a native English speaker through the editing service Enago (enago.com). We would greatly appreciate it if you could point out specific sentences or sections that could be simplified or improved. This feedback would allow us to address your suggestions effectively and also file a complaint with the company to ensure better quality in the future.
- Consider minor adjustments in sentence structure for complex or multi-clause sentences, particularly in sections with detailed statistical information, to improve readability.
Response: We would like to inform you that the article was initially reviewed by a native English speaker through the editing service Enago (enago.com). We would greatly appreciate it if you could point out specific sentences or sections that could be simplified or improved. This feedback would allow us to address your suggestions effectively and also file a complaint with the company to ensure better quality in the future.
- Your study addresses a novel topic with rigorous methods, yielding valuable insights for sports science and mental health. With minor adjustments, such as additional visual data and expanded discussion on broader applications, your work could make an even stronger impact in the field. Thank you for your contribution.
Response: Thank you for your positive and encouraging feedback. We are grateful for your recognition of the novelty and rigor of our study, as well as the valuable insights it provides for sports science and mental health. We have expanded the discussion on broader applications. We believe your suggestions have strengthened the impact of our work, and we have implemented them accordingly. Thank you again for your thoughtful contribution to improving our manuscript.
Reviewer 4 Report
Comments and Suggestions for Authors
1- The age range of the participants should be mentioned in the abstract.
2- More details about the intervention should be mentioned in the abstract.
3- A better conclusion should be written in the abstract. Now it is a bit vague and the reader cannot exactly understand the main finding.
4- On what basis was the sample size selected? Is statistical software used? Please explain in more detail in this context.
5- In Table 1, the effect size should be mentioned for all comparisons.
6- Was this study done in the COVID-19 pandemic? If your answer is yes, please mention it in this context and in the limitations section.
Author Response
Manuscript number: ijerph-3318686
Title: A Single Session of Beach Tennis Improves Anxiety Symptoms in Women But Not in Men: A Randomized Trial (former title: A Single Session of Beach Tennis Improves Anxious Symptoms in Healthy Women But Not in Healthy Men Recreational Players: A Within-Trial Randomized Clinical Trial)
Mia Sevic
Assistant Editor
International Journal of Environmental Research and Public Health
Dear Mia Sevic,
Thank you for the opportunity to resubmit our revised manuscript. We have carefully addressed all the reviewers’ comments and incorporated the requested revisions, clearly marked using MS Word’s track changes feature. Each reviewer’s question has been thoroughly addressed, and we believe these changes have significantly enhanced the quality of the manuscript. We are confident that the revised version aligns with the high publication standards of International Journal of Environmental Research and Public Health.
Response to Reviewer 4 Comments
- The age range of the participants should be mentioned in the abstract.
Response: Thank you for your suggestion. We have revised the abstract to include the mean age and standard deviation of the participants, addressing your request.
- More details about the intervention should be mentioned in the abstract.
Response: Thank you for your valuable feedback. In response to your request, we have provided additional details about the intervention in the abstract. Specifically, we included a description of the intervention sessions (one-on-one matches, two-on-two matches, and a control session). Please let us know if this information does not meet with your expectation.
- A better conclusion should be written in the abstract. Now it is a bit vague and the reader cannot exactly understand the main finding.
Response: Thank you for your feedback. We have revised the conclusion in the abstract to make it more specific and to clearly highlight the main findings of the study. The updated conclusion emphasizes the gender-specific anxiolytic effects observed and provides a clearer interpretation of the results.
- On what basis was the sample size selected? Is statistical software used? Please explain in more detail in this context.
Response: An “a priori” power calculation was not performed because no similar studies were found in the literature. On the other hand, the “post hoc” power calculations have been recently criticized in recent literature (see DOI:10.1002/gepi.22464). For these reasons, we included the effect sizes and the confidence intervals for a better interpretation and understanding of the true impact of the current results.
- In Table 1, the effect size should be mentioned for all comparisons.
Response: The effect size for all comparisons was included.
- Was this study done in the COVID-19 pandemic? If your answer is yes, please mention it in this context and in the limitations section.
Response: Thank you for your comment. This study was not conducted during the COVID-19 pandemic. Therefore, there is no need to mention it in the context or in the limitations section.
Round 2
Reviewer 1 Report
Comments and Suggestions for Authors
· The manuscript has been improved substantially.Still ,there are some major issues which have to be addressed
· The title should contain the term recreational athletes as this study cannot be generalized
· The data on Brazil is looks unnecessary and irrelevant in this study
· The inclusion criteria does not have any criteria to include only recreational athletes
· “discontinued participation during the study” should not be an inclusion criterion as participants will be included before data collection
· The authors didn’t calculate the sample size prior to the study.It is a major issue. If there is no adequate data for sample size calculation, the authors might have tried a pilot study for calculating sample size
· Table 3 is incomplete.It should contain actual p-value, effect size and confidence interval
Author Response
Manuscript number: ijerph-3318686
Title: A Single Session of Beach Tennis with Recreational Athletes Improves Anxiety Symptoms in Women but Not in Men: A Randomized Trial (former title: A Single Session of Beach Tennis Improves Anxiety Symptoms in Women But Not in Men: A Randomized Trial)
Mia Sevic
Assistant Editor
International Journal of Environmental Research and Public Health
Dear Mia Sevic,
Thank you for the opportunity to resubmit our revised manuscript. We have carefully reviewed all the comments from Reviewer 1, incorporating the requested revisions and clearly marking the changes using the MS Word track changes feature. Each point raised by the reviewer has been thoughtfully addressed, and we believe the improvements made have significantly enhanced the quality of the manuscript. We are confident that this revised version meets the high publication standards of the International Journal of Environmental Research and Public Health.
Thank you for your time and consideration. Please do not hesitate to contact us if further clarification or additional revisions are needed.
- The manuscript has been improved substantially. Still ,there are some major issues which have to be addressed
Response: Thank you for your positive feedback on the improvements made to the manuscript. We appreciate your valuable insights and have carefully revised the manuscript to address the remaining issues raised. Below, we provide detailed responses to each of your comments.
- The title should contain the term recreational athletes as this study cannot be generalized
Response: We greatly appreciate your valuable feedback regarding the title of our manuscript. We understand the importance of clarifying that the participants in the study are recreational athletes, and therefore, we have revised the title to properly reflect this information while still focusing on the results observed in both women and men.
The revised version of the title is as follows:
"A Single Session of Beach Tennis with Recreational Athletes Improves Anxiety Symptoms in Women but Not in Men: A Randomized Trial"
- The data on Brazil is looks unnecessary and irrelevant in this study
Response: Thank you for your comment regarding the inclusion of data related to Brazil. We understand your concern and have removed this information to ensure the study remains focused and relevant to a broader audience.
We appreciate your constructive feedback and remain available to make any further adjustments as needed.
- The inclusion criteria does not have any criteria to include only recreational athletes
Response: Thank you for your valuable comment. We acknowledge the importance of explicitly specifying the inclusion criteria for recreational athletes. In response, we have included the following information: the manuscript to include the following criterion in the inclusion section:
"(c) be recreational beach tennis players with at least 2 months of experience in beach tennis, ensuring a basic familiarity with the sport"
We appreciate your constructive input and believe this clarification strengthens the rigor of our methodology
- “discontinued participation during the study” should not be an inclusion criterion as participants will be included before data collection
Response: Thank you for pointing out this issue. We could not agree more with your observation, and for this reason, we have removed the expression “discontinued participation during the study” from the exclusion criteria.
We appreciate your careful review and constructive feedback, which has helped improve the clarity and accuracy of our manuscript. Please let us know is this change does not meet with you expectation.
- The authors didn’t calculate the sample size prior to the study. It is a major issue. If there is no adequate data for sample size calculation, the authors might have tried a pilot study for calculating sample size.
Response:
Thank you for your valuable comments. As we have answered to you in first round of revision, an “a priori” power calculation was not performed because no similar studies were found in the literature. Furthermore, we did not conduct a pilot study for calculating sample size. However, until present moment, there are only nine studies on beach tennis in PubMed. Of nine published studies, three are studies on injuries and one is a protocol suggestion. Of remain studies, the analysis of sample size indicated that our sample size is similar to published manuscript.
- Jung et al (2024) (doi: 10.3389/fphys.2024.1434636): The authors analyzed and compared the physiological responses of women during singles and doubles beach tennis sessions. To this end, a total of 22 female participants were recruited and randomly assigned to two groups.
- Ferrari et al (2024) (doi: 10.1097/HJH.0000000000003850): These authors determined the effect of 12 weeks of beach tennis training on 24-h ambulatory blood pressure in adults with essential hypertension. A total of 28 participants were submitted to beach tennis training.
- de Oliveira Carpes et al (2023) (doi: 10.3390/sports11030058): These authors evaluated the inter-individual BP responses after beach tennis, aerobic, resistance and combined exercise sessions in adults with hypertension. For beach tennis group, the authors included 23 participants.
- Domingues et al. (2022) (doi: 10.1097/MBP.0000000000000586): These authors investigated the acute effects of a single beach tennis session on short-term blood pressure variability in individuals with hypertension. To this end, 22 participants took part in this randomized clinical trial. They were randomly allocated to a beach tennis session and a nonexercise control session.
- Carpes et al (2021) (doi: 10.1007/s00421-021-04617-4): The authors evaluated the effect of a beach tennis session on 24-h ambulatory blood pressure in adults with hypertension. In this randomized crossover trial, 24 participants (12 men and 12 women) randomly performed two experimental sessions: a beach tennis session and a non-exercise control session.
To reinforce that small sample size was presented as study limitation to clarify.
- Table 3 is incomplete. It should contain actual p-value, effect size and confidence interval
Response: As requested, we have included p-value, effect size, and confidence interval to meet with your expectation.